# Soluble dimeric prion protein ligand activates Adgrg6 receptor but does not rescue early signs of demyelination in PrP-deficient mice

Anna Henzi[1], Assunta Senatore[1], Asvin K. K. Lakkaraju[1], Claudia Scheckel[1], Jonas Mühle[2], Regina Reimann[1], Silvia Sorce[1], Gebhard Schertler[2], Klaus V. Toyka[3], Adriano Aguzzi[1]*

1 Institute of Neuropathology, University of Zurich, Zurich, Switzerland, 2 Department of Biology and Chemistry, Paul Scherrer Institute, Villingen PSI, Switzerland, 3 Department of Neurology, University Hospital of Würzburg, University of Würzburg, Würzburg, Germany

* adriano.aguzzi@usz.ch

**Data Availability Statement:** The raw data from the RNA sequencing experiments are available on the GEO repository (accession number

## Abstract

The adhesion G-protein coupled receptor Adgrg6 (formerly Gpr126) is instrumental in the development, maintenance and repair of peripheral nervous system myelin. The prion protein (PrP) is a potent activator of Adgrg6 and could be used as a potential therapeutic agent in treating peripheral demyelinating and dysmyelinating diseases. We designed a dimeric Fc-fusion protein comprising the myelinotrophic domain of PrP ($FT_2Fc$), which activated Adgrg6 in vitro and exhibited favorable pharmacokinetic properties for in vivo treatment of peripheral neuropathies. While chronic $FT_2Fc$ treatment elicited specific transcriptomic changes in the sciatic nerves of PrP knockout mice, no amelioration of the early molecular signs demyelination was detected. Instead, RNA sequencing of sciatic nerves revealed downregulation of cytoskeletal and sarcomere genes, akin to the gene expression changes seen in myopathic skeletal muscle of PrP overexpressing mice. These results call for caution when devising myelinotrophic therapies based on PrP-derived Adgrg6 ligands. While our treatment approach was not successful, Adgrg6 remains an attractive therapeutic target to be addressed in other disease models or by using different biologically active Adgrg6 ligands.

## Introduction

The prion protein (PrP), encoded by the *PRNP* gene, is mainly known for its role as the causative infectious agent in prion diseases, a group of fatal neurodegenerative diseases. Yet the remarkable evolutionary conservation of PrP suggests that it exerts physiological functions. Mice ablated for PrP [1,2] and goats lacking PrP due to a naturally occurring mutation [3] develop a progressive peripheral demyelinating neuropathy, indicating that PrP is involved in myelin maintenance. Although no mutations in the human *PRNP* gene were found in a study of patients with hereditary neuropathies [4], the alteration of PrP or its sequestration in aggregates could explain the development of peripheral neuropathy in patients suffering from

GSE159948). All data and values that support the findings of this study are shown in Supporting Information S2 Table.

**Funding:** AH is the recipient of an MD PhD fellowship from Swiss National Foundation (project number 323530_171140). AA is the recipient of an Advanced Grant of the European Research Council, the Nomis Foundation and SystemsX.ch. AS and AKKL are recipients of grants from the Synapsis Foundation. KVT is the recipient of a senior professorship research grant by the University of Würzburg. AA and GS are the recipients of the Swiss National Foundation Sinergia grant CRSII5 183563. Swiss National Foundation: http://www.snf.ch/en/funding/Pages/default.aspx. The funders had no role in study design, data collection and analysis, decision to publish, or preparation of the manuscript.

**Competing interests:** The authors have declared that no competing interests exist.

Creutzfeldt-Jakob disease [5]. This notion is supported by the occurrence of pronounced peripheral demyelination in certain genetic forms of Creutzfeldt-Jakob disease [6]. Moreover, a patient with two pathogenic *PRNP* mutations was reported to develop an early onset peripheral demyelinating neuropathy [7].

The mechanism by which PrP exerts its function in myelin maintenance has recently been identified [8]. The N-terminal fragment, termed flexible tail (FT), comprises the myelinotrophic domain of PrP. FT is released by proteolysis and activates the adhesion G-protein coupled receptor Adgrgr6 on Schwann cells. Both in vitro and in vivo, activation of Adgrg6 by a peptide derived from FT results in cAMP accumulation and promyelinating signaling. In the peripheral nervous system (PNS), Adgrg6 is crucial for the development of the myelin sheath in zebrafish [9] and mice [10]. In addition, Adgrg6 is involved in the remyelination of axons [11] and reinnervation of neuromuscular junctions [12] after nerve injury. Whereas the inducible knockout of Adgrg6 in Schwann cells did not result in signs of demyelination for up to 4 months [11], aged conditional Adgrg6 knockout mice showed neuromuscular junction alterations and signs of denervation in hindlimbs, consistent with chronic disruption of Schwann cell function [12]. Together with the late-onset demyelinating neuropathy of PrP knockout mice, these findings suggest that Adgrg6 is not only required for the initiation of myelination, but also for long-term myelin maintenance.

The role of Adgrg6 in myelination and remyelination suggests that it could be a promising therapeutic target for peripheral demyelinating diseases and possibly other diseases linked to Adgrg6 malfunction, such as adolescent idiopathic scoliosis [13]. We therefore set out to explore the therapeutic potential of stimulating Adgrg6-dependent promyelinating pathways using its natural ligand PrP. To this end, we constructed a dimeric fusion protein consisting of the FT linked to crystallizable fragment (Fc) of immunoglobulin G1 ($FT_2Fc$). $FT_2Fc$ showed favorable pharmacokinetic properties in vivo including a half-life of 45 h but failed to have a therapeutic effect on the early molecular signs of demyelination in PrP knockout mice. Instead, gene expression analysis of sciatic nerves from mice treated with $FT_2Fc$ revealed unexpected changes in cytoskeletal and contractile elements. The observed transcriptomic changes were similar to the changes elicited by PrP overexpression in skeletal muscle, which causes a necrotizing myopathy [14,15].

## Material and methods

### Mice

Breeding and maintenance of mice was performed in specified-pathogen-free facilities at the University Hospital Zurich. Mice were housed in groups of 3–5, under a 12 h light/12 h dark cycle, with sterilized chow food and water *ad libitum*. The protocols for animal care and experiments were in accordance with the Swiss Animal Protection Law. All experiments were approved by the Veterinary Office of the Canton of Zurich (permits ZH139/2016 and ZH201/2018). For experiments with $FT_2Fc$, male and female C57BL/6J (WT) and PrP knockout (ZH3) mice were used. Mice were intravenously or intraperitoneally injected with various dosages of $FT_2Fc$, mIgG (5–10 mg/Kg bodyweight), sodium phosphate buffer (20 mM, pH 7), FT or control peptide. Blood samples were collected from the saphenous vein and serum was obtained by centrifugation of clotted blood for 1.5 min at 10'000 g. For the pharmacokinetic studies and acute treatment experiment, adult mice were used. Chronic treatment was started at 1 month of age and lasted for 4 months.

Transgenic mice expressing a ubiquitous promoter (CAG), a reporter gene-stop cassette (CAT) flanked by loxP sites, and *Prnp* were bred with mice expressing tamoxifen-inducible Cre recombinase under the control of the human ACTA1 (Actin, alpha 1, skeletal muscle)

promoter (C57BL/6J-Tg(CAG-cat,-Prnp)56Aag x Tg(ACTA1- cre/Esr1*)2Kesr/J). Tamoxifen-induced Cre-Lox recombination in skeletal muscle cells removed the stop-cassette and allowed for *Prnp* overexpression by the CAG promoter. Mice were fed food pellets with 400 mg/kg tamoxifen (Envigo) for one week to induce Cre-recombination. Male mice were sacrificed for organ collection 14 days after induction. CAG+/Cre+ mice were compared to CAG+/Cre-, CAG-/Cre+ and CAG-/Cre- mice. For collection of organs, mice were sacrificed by cervical dislocation in deep anesthesia.

### Electrophysiological investigations

Electrophysiological investigations were performed on the sciatic nerve of mice treated with $FT_2Fc$ or control treatment. Mice were anesthetized with Ketamine-Xylazine and electrophysiological investigations were performed as previously described [16,17]. The investigators were blinded as to the treatment and strain of the mice during the tests as well as post-hoc analyses.

### Morphological analysis

For toluidine blue stained sections, sciatic nerves were dissected and incubated in 4% glutaraldehyde in 0.1 M sodium phosphate buffer pH 7.4 at 4˚C overnight. Tissue was embedded in Epon using standard procedures and semithin sections (500 nm) were stained with toluidine blue. Counting of myelinated axons was performed manually on one 313 x 197 μM field of view per mouse with the observer blinded as to the treatment group. Then number of myelinated axons was normalized to the area. For cryosections of gastrocnemius muscle, the tissue was frozen in Optimal Cutting Temperature compound using liquid nitrogen and cut in 10 μM sections at the cryostat. Slides were incubated in 4% paraformaldehyde for 10 min and washed three times in PBS. The sections were first incubated in hematoxylin for 10 min, and then in trichrome staining solution (Chromotrope 2R 0.6% w/v (Chroma 1B259), fast green FCF 0.3% w/v (BDH 340304F), phosphotungstic acid 0.6% w/v, acetic acid 0.5%, pH 3.4) for 20 min. Finally, the sections were incubated in 0.5% acetic acid for differentiation and dehydrated in ethanol.

### Cell culture

Wild type SW10 cells and SW10 cells devoid of Adgrg6 were grown in DMEM (Gibco) supplemented with 10% FBS, penicillin-streptomycin and glutamax (Invitrogen) at 33˚C. FreeStyle™ 293-F (Thermofisher, R790-07) cells were grown in FreeStyle medium (Thermofisher, 12338018) on an orbital shaker (140 rpm) at 37˚C.

### Transfection

The mammalian expression vector for $FT_2Fc$ was obtained from ATUM (vector pD2610-v1). 293-F cells were seeded 2–3 h before transfection at a density of $1x10^6$ cells per ml in Freestyle medium supplemented with 0.2% w/v HyClone Cell Boost (GE Healthcare, SH30584.02). For transient transfection, 1 μg of plasmid and 2 μl of 40 kD linear polyethyleneimine (1 mg/ml, Polysciences, 24765–1) were used per milliliter of cells. For protein production, cells were supplemented with 2% w/v HyClone every 2–3 days (10% of culture volume each time). At 7 days post transfection, cell culture supernatant was cleared by centrifugation at 2'500 g for 10 min at 4˚C. The medium was sterile filtered and stored at 4˚C or –20˚C until further processing.

## Protein purification

Cell culture supernatant was diluted 1:1 in binding buffer (0.16 M sodium phosphate, 500 mM NaCl, pH 9) and loaded on a Protein A Sepharose (Sigma Aldrich, 17-1279-03) column at a flow rate of 2 ml/min. After washing with 5 column volumes of binding buffer until a stable baseline was reached, $FT_2Fc$ was eluted with elution buffer (0.1 M sodium citrate, pH 2.7). 500 μl fractions were collected and immediately neutralized with 50 μl 1 M Tris/HCl, pH 8. Elution fractions were analyzed by 12% or gradient 4–12% NuPAGE Bis-Tris gels (Invitrogen) followed by Coomassie staining. Fractions containing purified protein were pooled and dialyzed overnight against storage buffer (0.02 M sodium phosphate, pH 7) in a 10 kD cut off dialysis cassette (ThermoFisher, 66380). Purified protein was stored at 4˚C for use within days or at –20˚C for longer-term storage.

## RNA extraction, library preparation and sequencing

For sciatic nerves, total RNA was extracted using TRIzol (Invitrogen Life Technologies, 15596–018) according to the manufacturer's instructions. The phenol-chloroform phase was subsequently used for protein extraction. For RNA extraction of tibialis muscle from PrP over-expressing mice, the tissue was snap-frozen in liquid nitrogen and ground using the CryoGrinder™ kit system following the manufacturer's instructions. The RNA was extracted using the RNeasy Plus Universal Kit (Qiagen). Library preparation, RNA sequencing and bioinformatic analysis were performed at the Functional Genomics Center Zurich (FGCZ). For sciatic nerves of untreated 4 months old mice and tibialis muscle of PrP overexpressing mice, library preparation and RNA sequencing were performed as described previously [18]. For mice in the chronic treatment experiment and 13–15 months untreated old mice, libraries were prepared using the TruSeq RNA stranded Library Prep Kit (Illumina, lnc) and sequencing was performed on the Illumina Novaseq 6000 instrument for single-end 100 bp reads.

## RNA sequencing data analysis

For all samples, quality of reads was checked with FastQC. Reads were aligned to the GRCm38 genome assembly with Ensembl gene annotations with STAR [19]. Reads were counted with the featureCounts [20] function from the R package Rsubread. Differential expression analysis was performed with the R package EdgeR [21], using a generalized linear model with Trimmed Means of M-values (TMM) normalization. In the group pair-wise comparisons, we considered only genes with at least 10 raw counts in at least 50% of the samples in one of the groups. Genes with false discovery rate (FDR) below 0.05 were defined as differentially expressed. For the clustering analysis, the hclust function from the stats package was used. Visualizations were generated with the Sushi data analysis framework [22] provided by FGCZ or with R (version 3.5.2).

## Western blot analysis

SW10 cells were lysed in ice-cold lysis buffer (phosSTOP (Sigma, 4906845001) and protease inhibitor (Sigma, 11836153001) in RIPA buffer). Mouse organs were homogenized in lysis buffer using stainless steel beads. The lysates were centrifuged for 10 min at 10'000 g to remove debris. Protein concentration was measured with BCA assay (Thermo Scientific) and equal amounts of protein for each sample (10–30 μg unless otherwise noted) were boiled in 4 x LDS (Invitrogen) at 95˚C for 5 min. Mouse serum was diluted 1:10 in PBS and boiled in 4 x LDS with 0.1 M dithiothreitol (DTT) for western blotting. The samples were loaded on 12% or gradient 4–12% NuPAGE Bis-Tris gels (Invitrogen). Electrophoresis was performed at 200 V.

Gels were transferred to PVDF membranes with the iBlot system (Life Technologies). Membranes were blocked with 5% non-fat milk in PBS-T (for Fab83 staining) or 5% SureBlock (LuBioScience GmbH, SB232010) in TBS-T for all other staining. Then, membranes were incubated over night at 4˚C with the primary antibody diluted in blocking buffer. After three washes for 10 min, membranes were incubated with secondary antibodies coupled to horseradish peroxidase for 1 h at room temperature (RT). After washing, membranes were developed with Crescendo chemiluminescence substrate system (Sigma, WBLUR0500) and signals were detected using a Fusion Solo S imaging system (Vilber). Densitometry was performed using the FusionCapt Advance software. Dashed lines indicate removal of irrelevant lanes by image splicing from single blots. Original, uncropped images are shown in the Supporting Information (S1 Fig).

The following primary antibodies were used for western blotting: phospho-AKT (1:1000, Cell Signaling Technologies, 4060S), AKT (1:1000, Cell Signaling Technologies, 4685S), GFAP (1:2000, Cell Signaling Technologies, 12389S), c-Jun (1:1000, 9165s), Egr2 (1:2000, Abcam, ab108399), Calnexin (1:2000, Enzo, ADI-SPA-865-D), Actin (1:10'000, Milipore, MAB1501R). In addition, we used an in-house produced Fab-fragment directed against the N-terminus of PrP [23] (Fab83, 6 μg/ml). The following horseradish peroxidase coupled secondary antibodies were used: anti-mouse IgG (1:10'000, Jackson Immuno Research, 115-035-003), anti-rabbit IgG (1:4000, Jackson Immuno Research, 111-035-003), anti-human Fab (1:7000, Sigma, A0293).

## Enzyme-linked immunosorbent assay (ELISA)

384-well plates were coated with Fab83 (150 nM) or equimolar amount of BSA in PBS-T overnight at 4˚C. After three washes with PBS-T, plates were blocked with superblock (Thermofisher, 37515) for 2 h at RT. Next, a serial dilution of cell culture supernatant from transfected cells was incubated for 2 h at RT. After washing, goat anti-mouse IgG antibodies coupled to horseradish peroxidase (1:1000, Jackson Immuno Research, 115-035-003) were added for 1 h. Then, the plates were washed and developed with TMB (Invitrogen, SB02). The reaction was stopped with 0.5 M $H_2SO_4$ and absorbance at 450 nm was measured in a plate reader (Perkin Elmer, EnVision). The experiment was performed with technical duplicates.

## Immunoprecipitation

Immunoprecipitation (IP) of $FT_2Fc$ from cell culture supernatant was performed as previously described with minor modifications [23]. Briefly, sheep-anti mouse IgG paramagnetic beads (Dynal, 11201D) were coupled with anti-His mAb (Invitrogen, 37–2900) in coating buffer (PBS plus 0.1% immunoglobulin-free BSA) for 2 h at RT on a rotating wheel. Three molar excess of His-tagged Fab83 were added. After 1 h incubation, beads were washed three times with coating buffer. Cell culture supernatant was diluted 1:1 in IP buffer (50 mM Tris-Cl, 75 nM NaCl, 1% Igepal, protease inhibitor mixture (Sigma, 11836153001), pH 7.4) plus 0.5% BSA and incubated with 50 μl of Fab83-anti-His antibody coupled beads. The same input was used for all conditions. IP was performed overnight at 4˚C. After five washes with 50 mM Tris-Cl, 0.5% Igepal, 150 nM NaCl, 0.5% BSA, pH 7.4, elution of immunoprecipitated $FT_2Fc$ was performed by incubation with peptides (200 molar excess compared to Fab83) for 2 h at 4˚C. The eluate was boiled in 4 x LDS for western blotting. After elution, the beads were boiled in 4 x LDS and the supernatant was investigated by western blotting.

## Thermal shift assay

$FT_2Fc$ (1 μg) was diluted in 20 μl of 20 mM sodium phosphate, pH 7, with a final concentration of 10x SYPRO orange (Sigma, S5692). The temperature was increased from 25˚C to 95˚C at

3˚C per minute and fluorescence was measured at 610 nm in a Rotor-Gene Q thermocycler (Qiagen). The experiment was performed in technical triplicates. The fluorescence-temperature curves were fitted to the Boltzmann equation using GraphPad Prism (version 8.4.2) to determine the inflection point, which corresponds to the melting temperature.

### cAMP measurements

cAMP levels were measured as previously described [8] using a colorimetric competitive immunoassay (Enzo Life Sciences). Briefly, SW10 cells were plated in 6-well plates to ~50% density. Cells were incubated with $FT_2Fc$, FT or control treatment for 20 min and then lysed with 0.1 M HCl buffer for 20 min. The lysate was cleared by centrifugation at 600 g for 10 min and then processed according to the manufacturer's protocol.

### Experimental design and statistical analysis

GraphPad Prism software (version 8.4.2) was used for statistical analysis and data processing except for RNA sequencing data analysis. Normal distribution and equal variances of data were assumed, but this was not formally tested due to small *n* values. For the chronic treatment study, the sample size was determined with a power calculation based on the results from our previous electrophysiological studies [1]: at least 10 mice per group were required to detect a 6 m/s difference in motor nerve conduction velocity between treatment groups assuming a standard deviation of 4.4 m/s (power 90%, type I error 5%). For all other experiments, sample sizes were chosen according to sample sizes generally used in the field. For curve fitting, we used four-parameter logistic regression analysis or nonlinear least-squares analysis as indicated in the figure legends. As a measure for the goodness of fit we reported $R^2$ for nonlinear regression as computed by GraphPad Prism. Comparisons of two groups were performed by unpaired or paired two-tailed t-test as indicated. For comparison of three or more groups, one- or two-way ANOVA followed by multiple comparison tests were used and p-values were reported as multiplicity adjusted p-values. We used Sidak's multiple comparison test for comparison of preselected independent pairs, Bonferroni's multiple comparisons test when the assumption of independence could not be supported, and Dunnett's method for comparison of multiple groups to one control group. P-values < 0.05 were considered statistically significant. P-values are indicated in graphs as *: $p < 0.05$, **: $p = 0.01–0.05$, ***: $p = 0.001–0.01$, ****: $p < 0.0001$. ns = not significant, $p > 0.05$. Error bars in graphs show SEM. For in vitro experiments, individual points in the graphs correspond to independent experiments (cAMP assay, pAKT analysis) or technical replicates (ELISA, thermal shift assay). For animal experiments, each lane in the western blots and each point in the graphs corresponds to one mouse unless otherwise noted. For sciatic nerve protein analysis of chronically treated mice, one $FT_2Fc$ treated ZH3 mouse was excluded due a technical error in sample preparation. Otherwise, no samples or data were omitted during the analyses.

### Data availability

The raw data from the RNA sequencing experiments are available on the GEO repository (accession number GSE159948). All data and values that support the findings of this study are shown in Supporting Information S2 Table.

## Results

### Generation of a PrP-Fc-fusion protein

The binding of FT to Adgrg6 leads to intracellular accumulation of cAMP, which is essential in driving the synthesis of proteins and lipids critical for myelin generation and maintenance

[24]. We sought to investigate if sustained treatment with FT would suffice to constitutively activate Adgrg6 and thereby restore the reduced promyelination signaling in *Prnp* ablated mice (ZH3). Peptides are expected to have a short half-life in vivo, limiting their exposure to the target tissue. A common strategy to prolong the half-life of peptides is the fusion to larger molecules such as Fc [25]. We generated a mammalian expression plasmid containing FT (amino acids 1–50 of mouse PrP) fused with mouse IgG1-Fc at the hinge region, which upon transfection in cells leads to the expression of the homodimeric fusion protein, FT$_2$Fc (Fig 1A). The first 22 residues of FT comprise the signal peptide directing FT$_2$Fc for secretion, whereas residues 23–50 activate Adgrg6 [8]. Murine IgG1-Fc is unlikely to induce inadvertent activation of the immune system, since the murine IgG1 subclass binds to the inhibitory Fcγ-receptor and does not fix complement [26].

To assess whether FT$_2$Fc is correctly assembled and secreted, we transiently transfected FreeStyle 293-F cells with the plasmid expressing FT$_2$Fc, resulting in secretion of FT$_2$Fc into the culture medium. Under non-reducing conditions FT$_2$Fc predominantly existed as a dimer (56 kD), whereas in the presence of reducing agents (DTT) it migrated as a monomer (Fig 1B). We confirmed the identity of the secreted protein by western blotting using either anti-mouse IgG antibodies or a monomeric antibody Fab fragment specifically targeting the KKRPK domain of FT [23] (Fab83). Both antibodies detected FT$_2$Fc in cell culture supernatant as well as purified FT$_2$Fc (Fig 1B). To confirm the presence of FT$_2$Fc by additional methods, we performed a sandwich ELISA by capturing FT$_2$Fc with Fab83 and detecting the complex using anti-mouse IgG antibodies. A serial dilution of cell culture supernatant containing FT$_2$Fc resulted in a sigmoidal curve, whereas no signal was detected in medium from cells transfected with the empty vector (Fig 1C). Finally, we immunoprecipitated FT$_2$Fc from cell culture supernatant using beads coupled with Fab83. Peptides derived from the linear sequence of mouse PrP were previously used to map the epitope of Fab83 [23]. FT$_2$Fc could be eluted from the beads with a peptide competing for the Fab83 binding site (amino acids 23–34 of mouse PrP), but not with a non-competing peptide (amino acids 53–64) (Fig 1D).

Collectively, these results suggest that a correctly assembled FT$_2$Fc fusion protein was secreted by 293-F cells. We purified FT$_2$Fc from cell culture supernatant by protein A chromatography. The melting temperature of FT$_2$Fc as assessed by thermal shift assay was 74.5˚C at pH 7 (Fig 1E), which is similar to the melting temperature of other purified Fc fragments previously reported [27,28] and implies high thermal stability of FT$_2$Fc during storage and handling.

## FT$_2$Fc activates Adgrg6 signaling in vitro

cAMP signaling is involved in various steps of Schwann cell development (24) and is required for maintenance of a differentiated state [29]. Adgrg6 signals via cAMP and protein kinase A to initiate myelination in the PNS [30]. PI3K/AKT-signaling is another pathway involved in myelination and repair [31]. A link between cAMP and AKT activation has been suggested in vitro [32]. Activation of Adgrg6 by FT has previously been shown to induce cAMP signaling in primary Schwann cells, an immortalized Schwann cell line (SW10) and in vivo [8]. In addition, phosphorylation of AKT has been demonstrated in SW10 cells upon FT treatment [8].

We investigated whether purified FT$_2$Fc can act as a ligand for Adgrg6 in vitro. SW10 cells were incubated for 20 min with equinormal (half-equimolar where appropriate because of bivalency) amounts of either FT$_2$Fc (2.5 μM) or FT (amino acids 23–50 of mouse PrP, 5 μM). cAMP levels in the cell lysates were measured by ELISA as previously established [8]. Treatment of wild type SW10 cells (SW10$_{WT}$) with FT$_2$Fc resulted in a significant increase in the levels of cAMP, similar to that of FT treated cells. The increase in cAMP was not observed in Adgrg6-ablated cells (SW10$_{\Delta Adgrg6}$) (Fig 2A). Interestingly, we observed a bell-shaped dose-response-curve

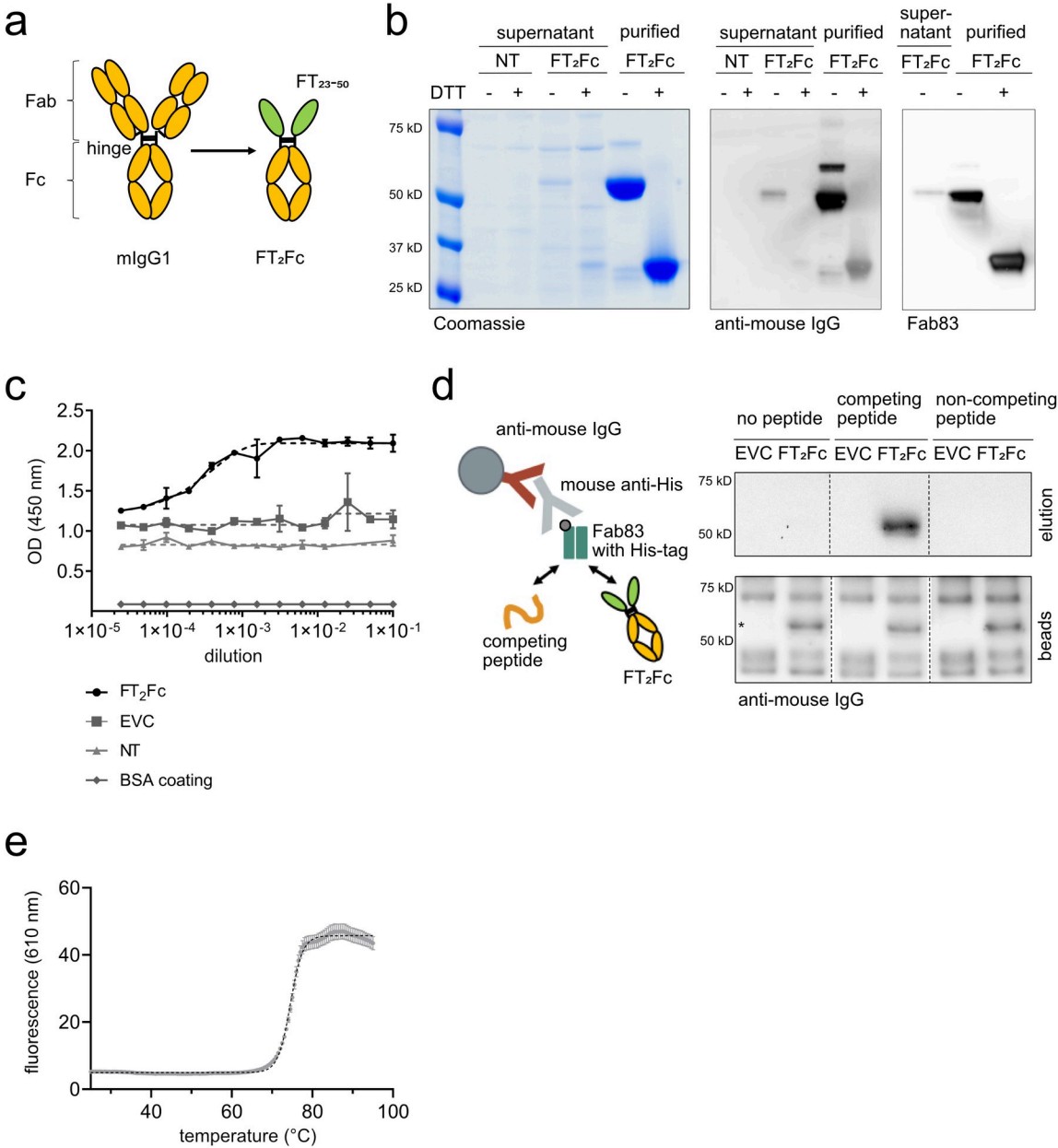

**Fig 1. Characterization of FT$_2$Fc.** (a) Design of FT$_2$Fc. FT was fused to mIgG1-Fc at the hinge, thereby replacing the antigen-binding fragment (Fab) and forming a homodimeric Fc-fusion protein. (b) FT$_2$Fc was secreted by 293-F cells after transient transfection and was present in the cell culture supernatant as a homodimer with a size of 56 kD. Under reducing conditions (+DTT), FT$_2$Fc was present as a monomer. FT$_2$Fc was detected in western blotting by antibodies targeting the Fc-fragment and a Fab specific for FT (Fab83). Supernatant from non-transfected cells (NT) was used as negative control. (c) Sandwich ELISA of serially diluted cell culture supernatant from cells transfected with FT$_2$Fc showed a sigmoidal curve. The optical-density (OD)—dilution curves were fitted using the four-parameter logistic nonlinear regression model (dashed lines, $R^2$ = 0.94 for FT$_2$Fc). Only background signal was detected in supernatant from cells transfected with the empty vector control (EVC) or non-transfected cells (NT), or when the plate was coated with BSA instead of Fab83. (d) Design of the immunoprecipitation assay (left). Western blots of eluates and beads boiled in sample loading buffer (right). FT$_2$Fc was captured in cell culture supernatant by beads coated with Fab83 and was eluted with a peptide specifically competing for the Fab83 binding site, but not with a non-competing peptide. The western blot from the beads confirmed that FT$_2$Fc (size of specific bands marked with $^*$) was bound to the beads in all conditions. (e) In the thermal shift assay, the unfolding of FT$_2$Fc with increasing temperature was monitored using a fluorescent dye. The curve was fitted to the Boltzmann equation (dashed line, $R^2$ = 0.99). The inflection point at 74.5°C corresponds to the melting temperature of FT$_2$Fc in 20 mM sodium phosphate, pH 7.

for $FT_2Fc$ in $SW10_{WT}$ cells with a maximum 9-fold increase in cAMP levels at 7 μM, and a drop in activity at higher doses of $FT_2Fc$ (Fig 2B). Bell-shaped dose-response curves have been described in the literature for ligands of GPCRs and may be related to the dimeric nature of $FT_2Fc$ or receptor desensitization [33,34]. Nonlinear regression analysis revealed an $EC_{50}$ of 3.49 μM, suggesting that for cAMP signaling in $SW10_{WT}$ cells, $FT_2Fc$ exhibited higher efficacy but slightly lower potency when compared to previously reported results for FT [8]. Then, we assessed the levels of phosphorylated AKT (pAKT) in SW10 cells upon $FT_2Fc$ treatment (Fig 2C). $SW10_{WT}$ cells, but not $SW10_{\Delta Adgrg6}$ cells showed a time-dependent increase in AKT phosphorylation when incubated with $FT_2Fc$. These results suggest that $FT_2Fc$ acted as a ligand of Adgrg6 and activated intracellular pathways similarly to FT.

## Establishing a functional read out for in vivo assay

In order to monitor if $FT_2Fc$ is functional in vivo, we injected ZH3 mice intravenously (i.v.) with either FT (600 μg) or $FT_2Fc$ (10 mg/Kg bodyweight). AKT phosphorylation was

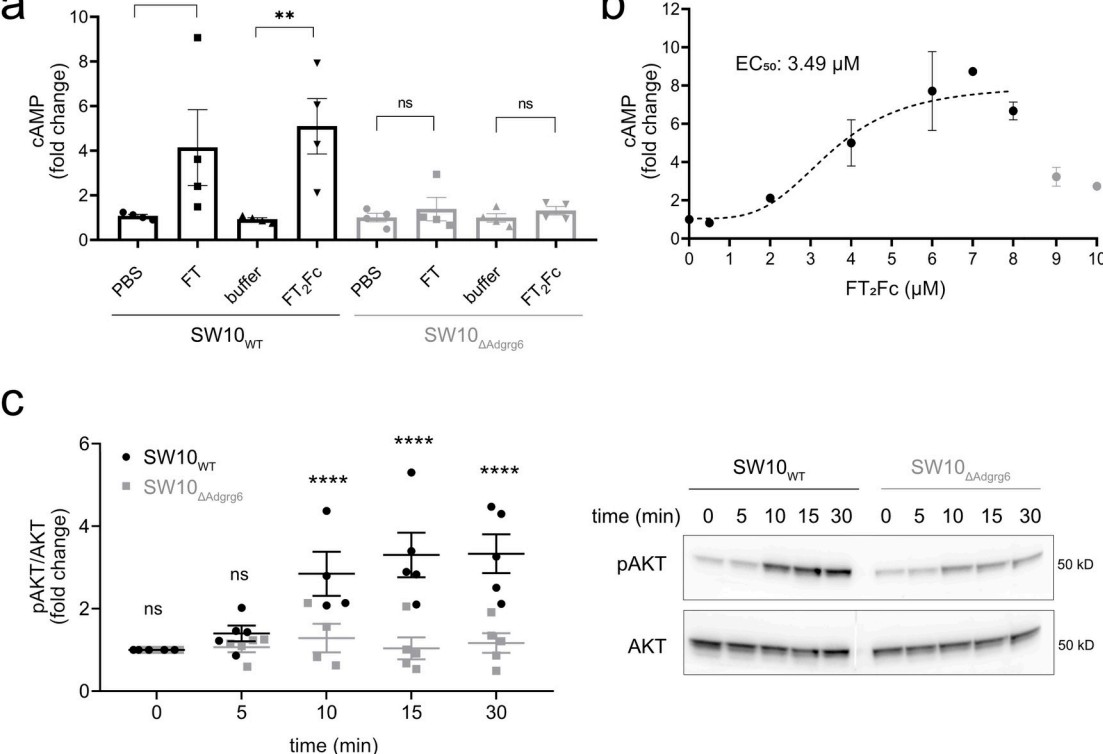

**Fig 2. Activation of Adgrg6 by $FT_2Fc$ in vitro.** (a) FT and $FT_2Fc$ elicited an increase in cAMP levels in $SW10_{WT}$ cells, but not in $SW10_{\Delta Adgrg6}$ cells. cAMP levels were measured 20 min after treatment with 5 μM FT or 2.5 μM $FT_2Fc$. To account for variability in cAMP levels across cell lines, cAMP was expressed as x-fold change to the average of the controls (PBS and 20 mM sodium phosphate buffer) for each of the 4 independent experiments. One-way ANOVA for selected comparisons, Sidak's multiple comparisons test: in $SW10_{WT}$ FT vs. PBS $p = 0.0392$, $FT_2Fc$ vs. buffer $p = 0.0035$. In $SW10_{\Delta Adgrg6}$ FT and $FT_2Fc$ vs. controls $p > 0.05$. (b) $SW10_{WT}$ cells were incubated with increasing concentrations of $FT_2Fc$. cAMP levels increased up to a maximum of a 9-fold change when compared to untreated cells. Nonlinear regression analysis revealed an $EC_{50}$ of 3.49 μM. For the curve fitting with the four-parameter logistic regression model (dashed line, $R^2 = 0.78$) only values from 0–8 μM (2–3 replicates per concentration) were used, because the strongly reduced activity at 9 and 10 μM would have confounded the analysis. (c) $SW10_{WT}$, but not $SW10_{\Delta Adgrg6}$ cells showed a time-dependent increase in AKT phosphorylation upon treatment with 1 μM $FT_2Fc$. Quantification of 5 independent experiments (left), representative western blot of cell lysates (right). Comparison of $SW10_{WT}$ to $SW10_{\Delta Adgrg6}$ cells with two-way ANOVA followed by Sidak's multiple comparisons test: at 10 min, 15 min and 30 min $p < 0.0001$, at 5 min and 10 min $p > 0.05$. ns = not significant.

measured in sciatic nerve lysates by western blotting (Fig 3A and 3B). FT injected mice showed a significant increase in the levels of pAKT after 30 minutes (relative pAKT change based on quantitative analysis of western blots: control $1.00 \pm 0.05(3)$; FT $1.53 \pm 0.11(3)$; $p = 0.0116$; mean $\pm$ SEM$(n)$; unpaired t-test), whereas FT$_2$Fc did not elicit an acute increase

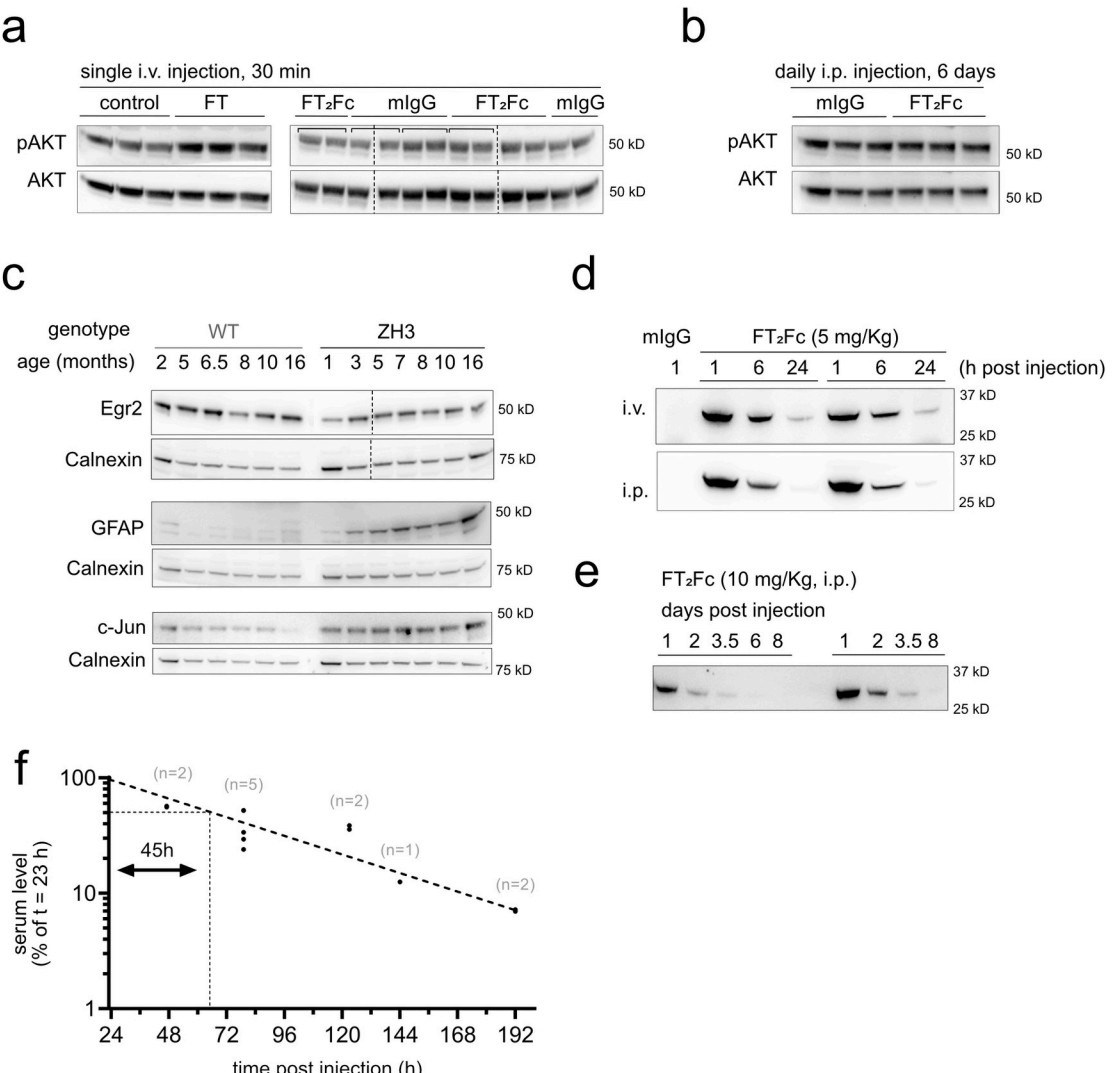

**Fig 3. Establishment of readout and pharmacokinetics.** (a) AKT phosphorylation increased in sciatic nerves of ZH3 mice 30 min after injection of 600 μg FT. As control (ctrl), mice were injected with an inactive peptide, in which lysine residues have been replaced with alanine residues [8]. In mice injected with FT$_2$Fc (10 mg/Kg), no significant change in AKT phosphorylation was detected after 30 min when compared to mIgG injected mice. Square brackets in western blots indicate left and right sciatic nerves taken from one mouse. The average value was used for quantification. (b) AKT phosphorylation did not increase after six consecutive FT$_2$Fc injections. (c) Western blots of sciatic nerve cell lysates from WT and ZH3 mice. ZH3 mice exhibited a decrease in sciatic nerve Egr2 levels and a concomitant increase in GFAP and c-Jun levels. (d) ZH3 mice were i.v. ($n = 2$) or i.p. ($n = 2$) injected with FT$_2$Fc (5 mg/Kg) and FT$_2$Fc serum levels were monitored by western blotting with Fab83 at 1, 6 and 24 h after injection. A serum sample from a mouse injected with mIgG (5 mg/Kg) was used as negative control. The early serum level–time course was similar in i.v. and i.p. injected mice. (e) ZH3 mice were i.p. injected with 10 mg/Kg FT$_2$Fc and blood samples were collected from 1 to 8 days post injection. The elimination of FT$_2$Fc from serum followed first order elimination kinetics. (f) Serum level values (normalized to the level at 23 h post injection) were plotted on a semi-logarithmic graph and fitted with nonlinear least-squares analysis (dashed line, $R^2 = 0.92$) to calculate the terminal serum half-life of FT$_2$Fc. The serum half-life was estimated to be 45 h with a 95% confidence interval of 37–58 h. The number of mice investigated per time point ($n$) is indicated above the points.

in AKT phosphorylation (control $1.00 \pm 0.04(4)$; $FT_2Fc$ $0.99 \pm 0.02(4)$; mean $\pm$ SEM$(n)$; $p = 0.8560$; unpaired t-test). Additionally, no significant increase in pAKT levels could be detected after 6 days of $FT_2Fc$ treatment (control $1.00 \pm 0.07(3)$; $FT_2Fc$ $1.14 \pm 0.02(3)$; mean $\pm$ SEM$(n)$; $p = 0.1356$; unpaired t-test). This result was not unexpected since $FT_2Fc$ is a larger molecule and predicted to diffuse at a slower rate to the target tissue as compared to FT. Accordingly, acute in vivo activity might be difficult to capture with our assay. We therefore proceeded with long-term $FT_2Fc$ treatment of ZH3 mice to assess therapeutic effects upon chronic exposure. As a read-out we aimed at identifying early protein expression changes. The transcription factor Egr2 (Krox20) is required for myelin maintenance [35] and has previously been shown to be decreased in ZH3 mice [8]. We collected sciatic nerves from ZH3 mice at various ages and found that Egr2 expression levels were lower when compared to wild type (WT) C57BL6/J mice (Fig 3C). The protein markers of repair Schwann cells, c-Jun and GFAP, showed an early increase in ZH3 mice when compared to WT mice. (Fig 3C). We planned to use these early signs of peripheral nerve damage as a readout to assess the effect of $FT_2Fc$ treatment in a chronic prophylactic treatment study.

## In vivo pharmacokinetics of $FT_2Fc$

To determine a suitable treatment regime for the in vivo experiments, we performed pharmacokinetic studies with $FT_2Fc$ in ZH3 mice. ZH3 mice were injected either i.v. or intraperitoneally (i.p.) with $FT_2Fc$ (5 or 10 mg/Kg). Blood samples were collected at different time points after a single injection (1 h up to 8 days) and $FT_2Fc$ levels were monitored in the serum by western blotting with Fab83. The distribution phase of $FT_2Fc$ was similar after 1 h post injection when comparing i.v. and i.p. injection (Fig 3D). Thus, we proceeded with i.p. injections for further experiments. In the elimination phase, $FT_2Fc$ followed first order elimination kinetics with an exponential decrease of serum levels over time (Fig 3E). Based on the serum level–time profile during the elimination phase the terminal serum half-life of $FT_2Fc$ was estimated to be 45 h (Fig 3F). This duration was deemed sufficiently long for chronic $FT_2Fc$ treatment in mice.

## Chronic treatment fails to rescue early molecular changes in sciatic nerves of ZH3 mice

We next examined if chronic administration of $FT_2Fc$ in ZH3 mice would rescue the demyelinating neuropathy. We designed a prophylactic study in which treatment was started at 1 month of age, when signs of demyelination are not yet present in ZH3 mice [1,36] (Fig 4A). ZH3 mice were injected i.p. three times per week (based on the 45 h half-life of $FT_2Fc$) with 8 mg/Kg $FT_2Fc$ or control treatment for a total of 4 months. We additionally included WT mice, which received the same treatment as ZH3 mice. Blood samples were collected at 1 and 2 months after treatment start. Serum levels of $FT_2Fc$ were comparable at both time points (Fig 4B), indicating that the half-life of $FT_2Fc$ did not decrease over time and that mice did not generate antibodies against $FT_2Fc$ [37,38]. Furthermore, we detected $FT_2Fc$ in sciatic nerve homogenate by western blotting (S2 Fig). There was no difference in body weight between the treatment groups (Fig 4C). At the end of the chronic treatment mice were sacrificed and sciatic nerves were collected for protein analysis (Fig 4D). GFAP levels were increased whereas Egr2 levels were decreased in sciatic nerves of ZH3 mice when compared to WT mice. However, treatment with $FT_2Fc$ did not rescue the increase of GFAP nor the decrease in Egr2 levels. C-Jun levels were not increased in ZH3 mice when compared to WT mice, nor did $FT_2Fc$ treatment change c-Jun levels. As expected at the age of 5 months, sciatic nerves of ZH3 mice

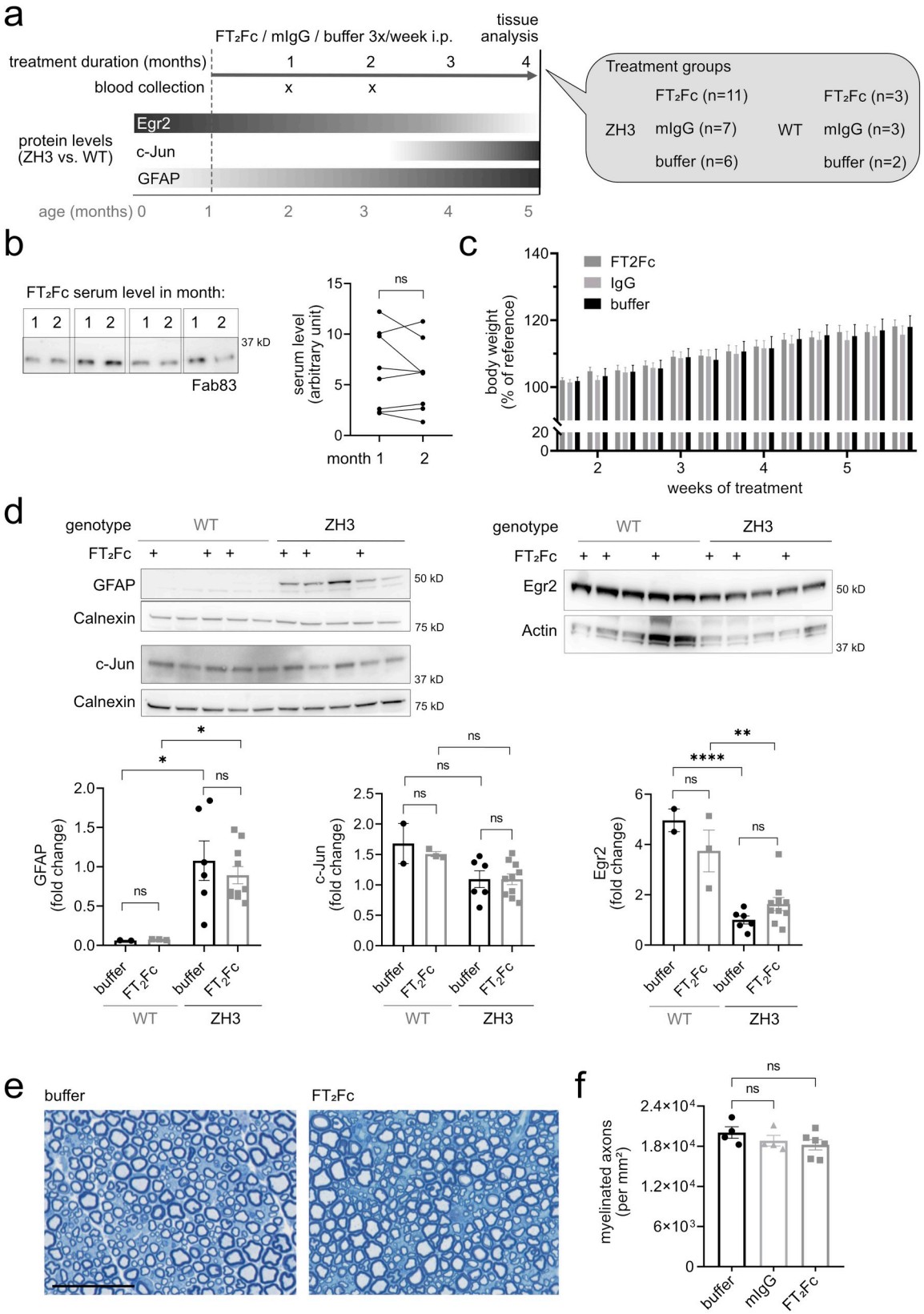

**Fig 4. Chronic administration of FT$_2$Fc.** (a) Design of the prophylactic treatment experiment in ZH3 and WT mice. Starting at 1 month of age, ZH3 and WT mice were injected with FT$_2$Fc (or control treatment) 3 times per week until the age of 5 months. Bars with grey shades show expected changes in protein levels with dark and light grey meaning high and low levels, respectively. (b) FT$_2$Fc serum levels were measured 2 days after the last injection. Representative western blot showing serum levels for 4 ZH3 mice after 1 and 2 months of treatment. FT$_2$Fc serum levels did not decrease after 2 months of treatment when compared to the level at 1 month ($n = 8$, paired t-test, $p = 0.3529$), suggesting that FT$_2$Fc half-life was unaltered. (c) Bodyweight was not significantly different between treatment groups (as analysed by two-way repeated measures ANOVA, $p > 0.05$). Body weight was recorded at every injection time point for all the mice in the chronic treatment study and is here shown for week 1 to 6 of treatment as percentage of the body weight at treatment start (reference). (d) Representative western blots and quantification for levels of GFAP, c-Jun and Egr2 in sciatic nerves. GFAP levels were significantly higher in ZH3 mice compared to WT mice, but no significant change was induced by FT$_2$Fc compared to buffer treatment (one-way ANOVA for selected comparisons, Bonferroni's multiple comparisons test: ZH3 vs. WT for buffer treated mice $p = 0.0345$, for FT$_2$Fc treated mice $p = 0.0343$. FT$_2$Fc vs. buffer treated mice, both ZH3 and WT $p > 0.05$). C-Jun levels were not decreased in ZH3 compared to WT mice, and no significant change in c-Jun levels was induced by FT$_2$Fc treatment (one-way ANOVA for selected comparisons, Bonferroni's multiple comparisons test, $p > 0.05$). Egr2 levels were significantly lower in ZH3 mice compared to WT mice, but FT$_2$Fc treatment did not induce a significant change in any genotype (one-way ANOVA for selected comparisons, Bonferroni's multiple comparisons test: ZH3 vs. WT for buffer treated mice $p < 0.0001$, for FT$_2$Fc treated mice $p = 0.0044$. FT$_2$Fc vs. buffer treated mice, both ZH3 and WT $p > 0.05$). Protein levels were expressed as fold change the average level in buffer treated ZH3 mice. (e) Toluidine-blue stained semithin sections of sciatic nerves. No difference in fibre morphology was detected between treatment groups in our ZH3 mouse specimens. Scale bar 50 µM. (f) Number of myelinated axons per mm$^2$ in sciatic nerves of ZH3 mice revealed no difference between treatment groups (comparison of FT$_2$Fc and IgG treated to buffer treated mice by one-way ANOVA followed by Dunnett's multiple comparisons test, $p > 0.05$). ns = not significant.

showed no conspicuous morphological signs of demyelination [1,36], and no morphological alterations were induced by FT$_2$Fc treatment (Fig 4E and 4F).

Certain phenotypes in early generations of PrP knockout mice were found to be poorly reproducible and seem to represent genetic confounders [36,39]. The development of peripheral demyelination, however, was confirmed in ZH3 mice that have a pure C57BL6/J background [36]. Based on the time course of macrophage infiltrations, the disease seemed to manifest later than described in previous reports on mice with mixed genetic background [1]. The early reduction in nerve conduction velocity has not been re-assessed in ZH3 mice. In electrophysiological investigations performed at the end of FT$_2$Fc treatment, we did not detect a decrease in nerve conduction velocity in ZH3 mice when compared to WT mice (Fig 5A and 5B). This suggests that the ZH3 mice used in the experiments reported here had not developed electrophysiological signs of demyelination at the age of 5 months and that these investigations were not a telling readout for a treatment effect in our study. Indeed, we could not detect any differences in nerve conduction velocity or compound muscle action potential amplitude between treatment groups (Fig 5A–5C). No polyphasic compound muscle action potentials were recorded, and electromyography of the foot muscles showed no pathological spontaneous activity.

## FT$_2$Fc treatment elicits gene expression changes which are deleterious in skeletal muscle

In the absence of a rescue of protein markers in ZH3 mice upon chronic FT$_2$Fc treatment, we postulated that FT$_2$Fc either did not reach the precise destination in the sciatic nerve or did not activate the desired myelinotrophic signalling pathways. To assess FT$_2$Fc induced changes in an unbiased and genome-wide manner, we investigated the transcriptome of sciatic nerves from FT$_2$Fc treated and control mice by RNA sequencing. Additionally, we sequenced sciatic nerves of untreated WT and ZH3 mice at 4 and 13–15 months.

Unsupervised clustering based on the 100 genes with the highest variance across all samples showed a separation between FT$_2$Fc treated and control treated ZH3 mice (Fig 6A). Based on published datasets, we assembled a list of genes that are important for the repair Schwann cell phenotype [40]. ZH3 mice showed a mild increase of these genes at 4 months and a strong upregulation at 13–15 months (Fig 6B). We did not observe a change in expression of any of

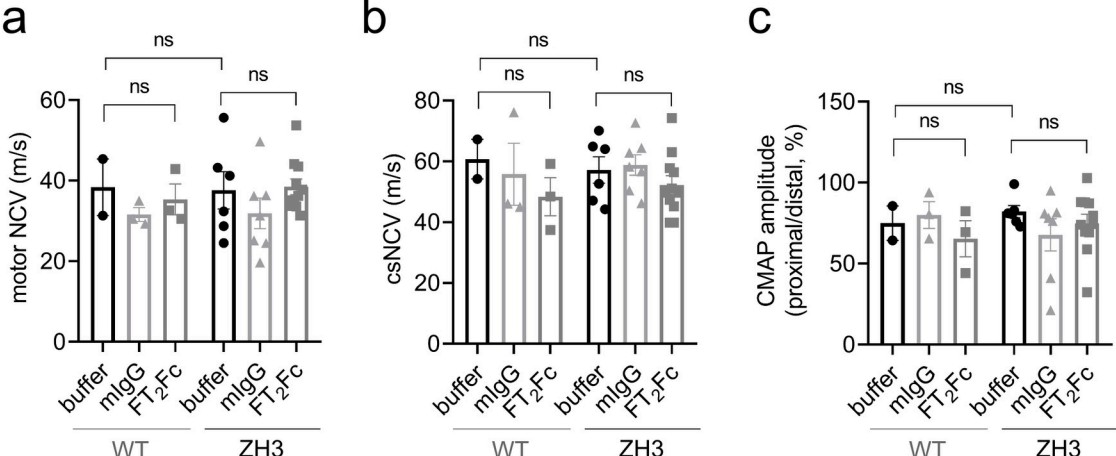

**Fig 5. Electrophysiological studies.** All tests and calculations were done with the examiners masked as to treatment and strain allocation. (a) Motor nerve conduction velocities (NCV) for WT and ZH3 mice. There was no significant difference between WT and ZH3 mice, or between $FT_2Fc$ treated and buffer treated mice (one-way ANOVA for indicated comparisons, Bonferroni's multiple comparisons test, $p > 0.05$). (b) Compound sensory NCV (csNCV) for WT and ZH3 mice. Again, there was no significant difference between WT and ZH3 mice, or $FT_2Fc$ treated and buffer treated mice (one-way ANOVA for indicated comparisons, Bonferroni's multiple comparisons test, $p > 0.05$). (c) The ratio of proximal to distal compound muscle action potential (CMAP) amplitude was not significantly different when comparing ZH3 to WT mice or $FT_2Fc$ to buffer treated mice (one-way ANOVA for indicated comparisons, Bonferroni's multiple comparisons test, $p > 0.05$). ns = not significant.

these genes upon $FT_2Fc$ treatment (Fig 6B), which is in line with the absence of changes in repair Schwann cell markers at the protein level. Instead, functional gene ontology analysis revealed that $FT_2Fc$ specifically induced a downregulation of genes associated with muscle contraction and organization of actin filaments and sarcomeres (Table 1 and Fig 6C). Our sciatic nerve bulk RNA sequencing analysis did not allow us to infer in which cell type these genes were differentially expressed. We therefore explored previously published single cell RNA sequencing data from healthy and injured sciatic nerves [41]. Several genes that were differentially expressed upon $FT_2Fc$ treatment are enriched in perivascular cells and vascular smooth muscle cells (Des, Sh3bgr, Tpm1, Ldb3) or belong to mesenchymal cell clusters (Tnnt3, Cmya5, Pygm, Eno3). In contrast, genes that are expressed specifically in Schwann cells did not change their expression profile upon $FT_2Fc$ treatment.

These unexpected experimental observations led us to consider previous studies that showed a necrotizing myopathy in mice overexpressing PrP [14,15]. We wondered whether treatment with FT2Fc may have mimicked PrP overexpression and induced gene expression changes similar to those involved in the myopathy phenotype rather than acting as a ligand to Adgrg6. We therefore compared the transcriptomic changes in $FT_2Fc$ treated sciatic nerves to gene expression changes in the tibialis anterior muscle upon inducible PrP overexpression (Fig 7A and 7B). While the overall correlation between the datasets was low, a noteworthy overlap was observed among the downregulated genes. Specifically, 35% of genes that were significantly downregulated (FDR < 0.05) by $FT_2Fc$ were also downregulated in PrP overexpressing muscle. This suggests that $FT_2Fc$ treatment might have triggered pathways similar to those caused by PrP overexpression. However, the muscle tissue of $FT_2Fc$-treated mice did not show any myopathic changes (Fig 7C).

In conclusion, the transcriptomic analysis showed that chronically administered $FT_2Fc$ had reached the sciatic nerve and elicited a specific pharmacodynamic effect. Yet the detected gene expression changes did not indicate an activation of Adgrg6-mediated myelination signalling.

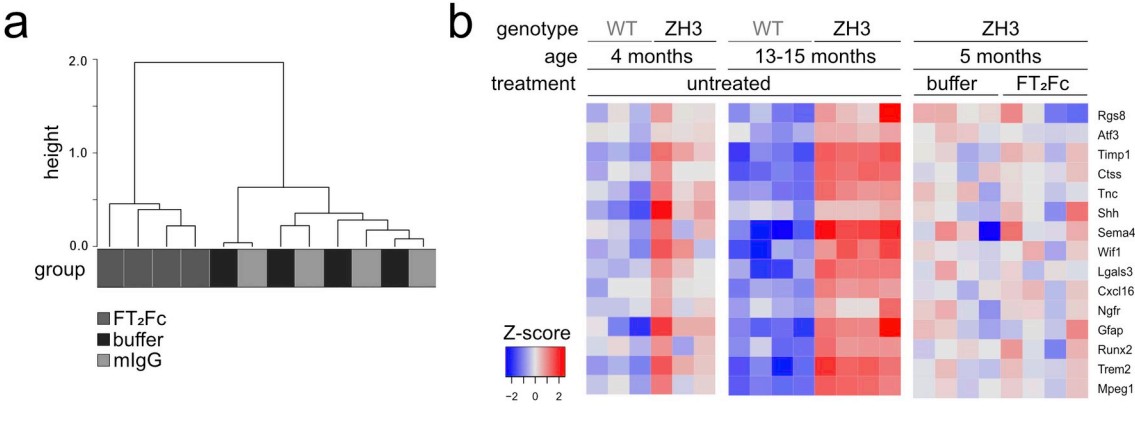

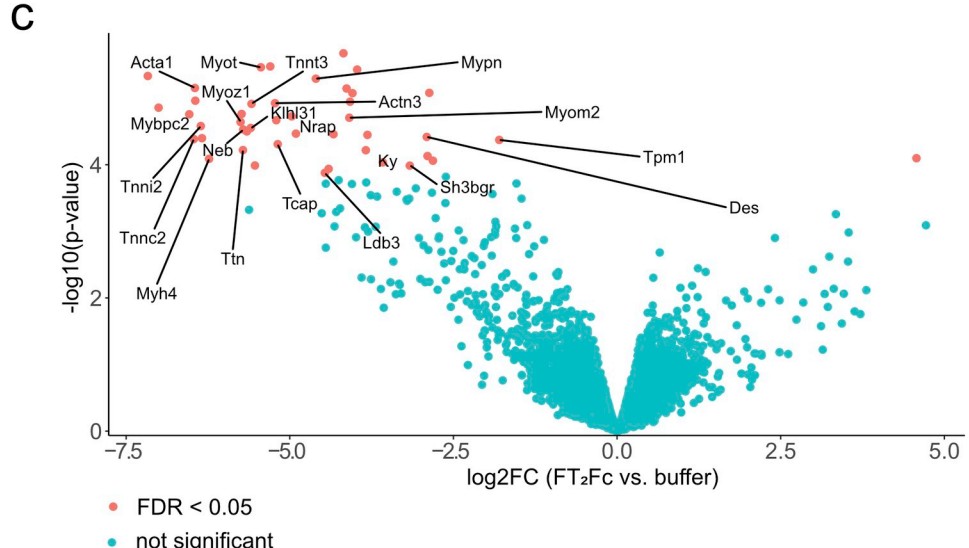

• FDR < 0.05

• not significant

**Fig 6. RNA sequencing of sciatic nerves from FT₂Fc treated mice.** (a) Hierarchical clustering analysis based on the 100 genes with the highest variance across all samples showed a separation between FT₂Fc treated mice and control mice (n = 4 per treatment). (b) Heatmap of the selected repair Schwann cell genes. The red-blue colour key is based on the row-wise Z-scores. 4 months old ZH3 mice (n = 3) showed a mild, 13–15 months old ZH3 mice (n = 4) a pronounced upregulation of these genes when compared to age-matched WT mice. No difference was detected in FT₂Fc treated ZH3 mice (n = 4) compared to buffer treated mice (n = 4). (c) Volcano plot showing differentially expressed genes in sciatic nerves of FT₂Fc treated compared to buffer treated mice. Genes with FDR < 0.05 were considered significantly up- or downregulated (43 downregulated genes, 1 upregulated gene). Genes involved in actin binding are labelled in the plot. Log2FC = log2 fold change.

Instead, the downregulation of cytoskeleton proteins and contractile elements was reminiscent of the toxic effects of PrP overexpression.

## Discussion

The treatment options for many peripheral nerve diseases are limited, and despite the remarkable ability for repair in the PNS, regeneration in intrinsic peripheral neuropathies or after nerve injury is often incomplete. Schwann cells are crucial for the function and maintenance of peripheral nerves and may represent interesting targets to boost the endogenous repair capacity of the PNS.

Many approved drugs target G-protein coupled receptors, and therefore Adgrg6 may represent an attractive potential target to stimulate repair in peripheral neuropathies or after nerve

**Table 1. Biological process (BP) enriched categories obtained by gene ontology analysis.**

| BP enriched category | Downregulated genes | P adjusted | List of genes |
|---|---|---|---|
| Muscle contraction | 18 | < 0.00001 | Tmod4; Actn3; Trdn; Mybpc1; Myom1; Cacna1s; Tpm2; Trim63; Ryr1; Myom2; Myh2; Mybpc2; Ttn; Actn2; Myh7; Myh1; Myh4; Myl1 |
| Sarcomere organization | 13 | < 0.00001 | Casq1; Tcap; Mybpc1; Mypn; Ldb3; Myom1; Neb; Myom2; Mybpc2; Mybph; Ttn; Actn2; Klhl41 |
| Actin filament organization | 10 | < 0.00001 | Tmod4; Mybpc1; Myom1; Tpm2; Myom2; Pdlim3; Tpm1; Mybpc2; Mybph; Ttn |
| Glycogen metabolic process | 5 | 0.001 | Phkg1; Pygm; Agl; Phka1; Ppp1r3a |
| BP enriched category | Upregulated genes | P adjusted | List of genes |
| positive regulation of B cell activation | 6 | < 0.00001 | Ighv4-1; Ighv7-1; Ighv1-80; Ighv1-82; Ighv2-9-1; Ighv10-1 |
| immunoglobulin production | 3 | 0.02 | Igkv17-121; Igkv2-112; Igkv1-110 |

Gene ontology analysis of differentially expressed genes in sciatic nerves of FT$_2$Fc treated compared to buffer treated mice. The complete list of overrepresented categories is shown in S1 Table.

damage. We here exploited a natural Adgrg6 agonist, PrP, to design a molecule targeting Adgrg6 for the treatment of peripheral nerve disease. The fusion of PrP's myelinotrophic domain to an Fc-fragment (FT$_2$Fc) yielded a molecule with sufficiently long half-life, making it feasible for in vivo treatment. Several Fc-fusion based drugs (also termed immunoadhesins) have been approved for therapeutic use in humans [42]. Importantly, immunoadhesins are intrinsically dimeric. In the case of sufficiently dense targets, this allows for two-point binding. The theoretical avidity of such binding corresponds to the product of the monomeric binding affinity and can be extremely high [43,44].

We attempted a proof-of-principle study by prophylactically administering FT$_2$Fc to ZH3 mice, which develop a slowly progressive peripheral demyelinating neuropathy during their lifetime [1]. However, FT$_2$Fc did not activate the desired pro-myelination signaling pathways in our treatment study and we failed to detect a therapeutic effect in vivo. Instead, we detected a downregulation of cytoskeleton-related genes, reminiscent of the changes seen in skeletal muscle of PrP overexpressing mice developing a necrotizing myopathy [14]. While comparison of the transcriptional changes in different tissues as was presented here must be interpreted with great caution, the possibility of myotoxicity induced by a PrP-based agent should not be discounted. Early studies suggested that Adgrg6 is expressed in muscle tissue [45], but a recent study objected that Adgrg6 detected in muscle homogenate derived from endothelial cells instead of muscle cells [46]. Thus, it remains unclear whether the toxic effect of PrP is mediated by Adgrg6 or by other interaction partners.

Myelination and the maintenance of myelin are controlled by tightly balanced reciprocal signalling between Schwann cells, axons and extracellular matrix [47]. Inappropriate or excessive activation of involved signalling pathways can have detrimental effects on myelin development, maintenance or repair [48,49]. In our chronic treatment study, FT$_2$Fc did not negatively affect myelin maintenance as assessed by biochemical, morphological and electrophysiological investigations.

## Limitations of this study

Several reasons may explain the lack of a demonstrable treatment effect of FT$_2$Fc in vivo. Peripheral nerve damage in ZH3 mice at the age investigated here was only mild, as indicated by the lack of electrophysiological and morphological signs of demyelination. But even at the molecular level, no amelioration of the early protein and RNA changes was achieved by chronic

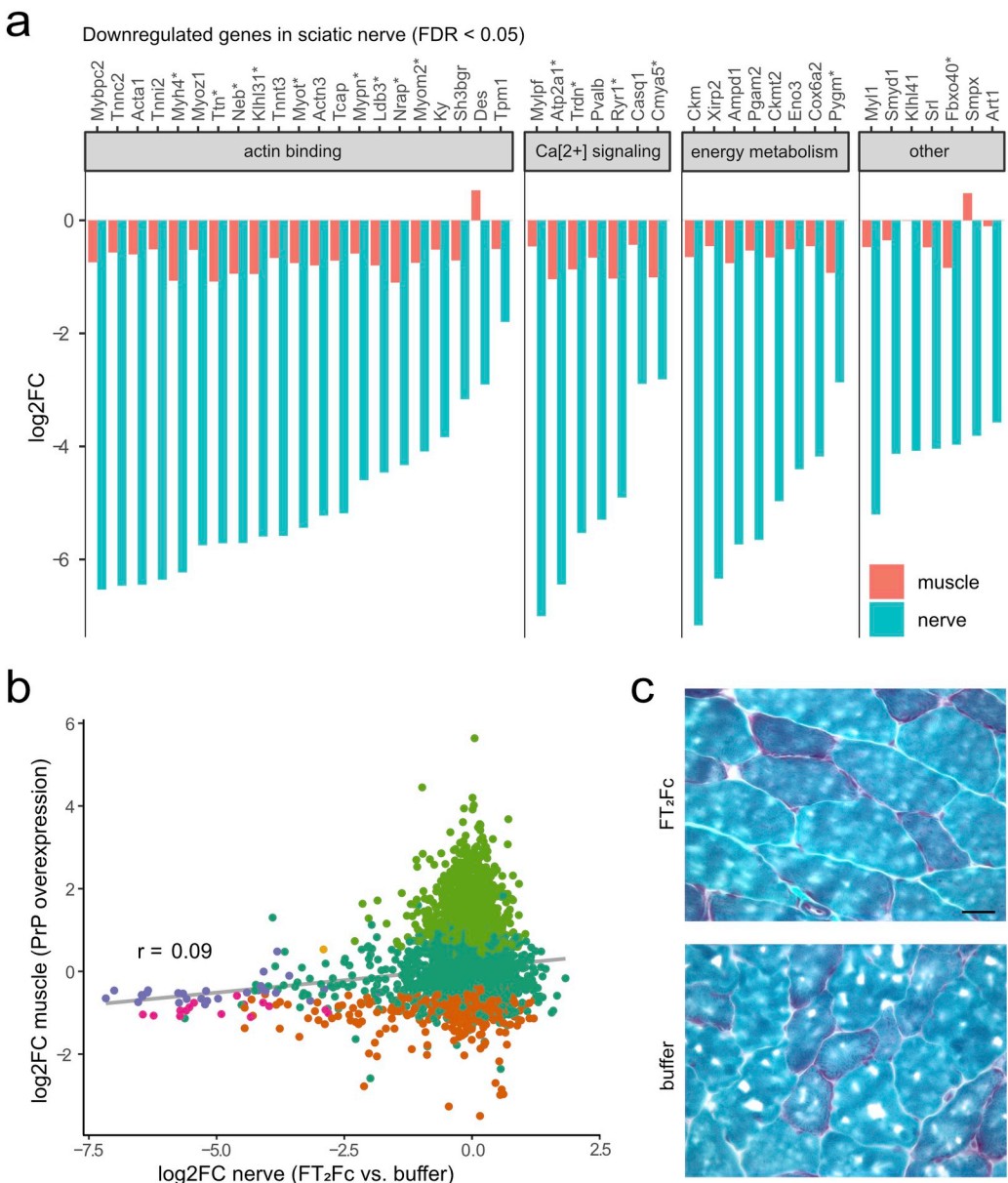

**Fig 7. Genes downregulated by FT$_2$Fc show similar changes in PrP-overexpressing muscle.** (a) Log2 fold change (log2FC) for significantly downregulated genes ($n = 43$, based FDR < 0.05) in sciatic nerves of FT$_2$Fc treated compared to buffer treated mice and corresponding log2FC in tibialis anterior muscle of PrP overexpressing mice. * marks genes which were also significantly downregulated in muscle based on FDR < 0.05 ($n = 15$). Grouping of genes according to common functions was based on a manual search in the UniProt database. (b) Scatterplot comparing gene expression changes in muscle of PrP overexpressing mice and sciatic nerves of FT$_2$Fc treated mice. 1229 and 43 genes were significantly (FDR < 0.05) downregulated in muscle and nerve, respectively, with 15 of these genes overlapping. Grey line represents linear regression. Pearson's correlation coefficient r = 0.09. (c) 10 μM Gomori trichrome stained frozen sections of gastrocnemius muscle from FT$_2$Fc and buffer treated ZH3 mice. Muscle fibre morphology was similar in buffer and FT$_2$Fc treated mice. No myopathic changes were detected in FT$_2$Fc treated mice. Scale bar 20 μM.

FT$_2$Fc treatment. We wondered whether this may have been caused by disadvantageous phar-macokinetic properties of FT$_2$Fc. We detected FT$_2$Fc in sciatic nerve lysate by western blotting, but this did not allow us to conclude that FT$_2$Fc reached Adgrg6 on Schwann cells. It is possible that, as a relatively large molecule, FT$_2$Fc might not have reached Adgrg6 in sufficient

quantities. However immunoglobulins, which are three times larger than FT$_2$Fc, were reported to cross the blood-nerve-barrier [50]. Alternatively, an uneven distribution caused for example by binding of FT$_2$Fc to Fcγ-receptors in various tissues could have sequestered FT$_2$Fc from the peripheral nerves. Reassuringly, the transcriptomic analysis revealed specific gene expression changes in the sciatic nerves of FT$_2$Fc treated mice, suggesting that FT$_2$Fc reached its destination and was pharmacodynamically active. Yet FT$_2$Fc did not activate the desired signaling pathways. The dimeric nature of FT$_2$Fc could alter its biological activity when compared to monomeric FT and endogenous PrP. A soluble dimeric full-length PrP was previously shown to have different properties that endogenous PrP with regard to its pathologic structural conversion [51]. Whether the dimeric full-length PrP is able to sustain its physiological function in myelin homeostasis has not been investigated by Meier et al. Moreover, oligomerization of receptors including adhesion G-protein coupled receptors, contributes to biased signaling [52,53]. A dimeric ligand such as FT$_2$Fc may exhibit novel binding properties, induce receptor homo- or hetero-oligomerization and activate different intracellular signaling pathways in vivo. Our in vitro studies with FT$_2$Fc showed increased cAMP elevation and prolonged AKT phosphorylation when compared to FT. While we interpreted these results as evidence for a high potency and stability of FT$_2$Fc, they could also indicate biased agonism and contribute to the unexpected pharmacodynamics properties of FT$_2$Fc in vivo [54]. Interestingly the bell-shaped dose-response curve observed here was not seen when cells were treated with FT [8] and could indicate a novel activity of dimeric FT$_2$Fc different from that of monomeric FT.

## Conclusions

To the best of our knowledge, this study represents the first attempt to target Adgrg6 activation for treating peripheral nerve demyelination in vivo. Although our treatment regimen was not successful, Adgrg6 should not be discarded as a possible therapeutic target for peripheral nerve diseases. For example, FT$_2$Fc could be tested in more rapidly progressive models of peripheral demyelination such as autoimmune peripheral neuropathies [55]. Moreover, the pharmacokinetics and pharmacodynamics of Adgrg6 ligands might be optimized by modifications such as coupling FT to polyethylene glycol or dendrimers or by designing a bispecific molecule targeting Adgrg6 and thereby directing FT to its receptor. Such strategies may be hampered by the blood-nerve barrier, which may hinder access of the therapeutic compound to Adgrg6 on Schwann cells. Encouragingly, two studies have identified novel agonists of Adgrg6 by drug screening in zebrafish [56,57]. Such small-molecule drugs may exhibit better penetration into nerves than the biological macromolecule studied here and could be leveraged to target Adgrg6 in vivo. Future studies should explore these alternative therapeutic strategies utilizing a variety of disease models.

## Supporting information

**S1 Fig. Original blots and gels from all figures.** The uncropped images have been inverted and autoscaled using the Quantify One software (Biorad). Lanes with protein size markers (Protein Precision Plus, Biorad) are indicated with M. Relevant size markers are marked with the respective size in kD. The specific band is marked with * when additional non-specific bands are present. Irrelevant lanes that have been excluded from the main figures are marked with x. The samples that were loaded in these lines are described in annotations. Lanes that were left empty are marked with e. The samples loaded in all other lanes were specified in the main figures.
(PDF)

**S2 Fig. Detection of FT$_2$Fc in tissue homogenates.** Tissue lysates from a mouse chronically treated with FT$_2$Fc were subjected to western blotting with Fab83. FT$_2$Fc was detected in all organs (100 μg of total protein) and in the serum (diluted 1:100 in PBS). Calnexin was used as a loading control. The mouse was sacrificed 2 days after the last injection. \*\*: dimer. \*: monomer.
(TIF)

**S1 Table. Complete list of overrepresented GO categories in sciatic nerves of FT$_2$Fc treated compared to buffer treated mice.**
(DOCX)

**S2 Table. Raw data and values used in the study.**
(XLSX)

# Acknowledgments

We thank Mirzet Delic, Ezio Luongo and Olga Romashkina for technical help with mouse experiments; Rita Moos and Simone Hornemann for advice and technical help regarding protein purification; FGCZ staff (specifically Maria Domenica Moccia and Giancarlo Russo) for generation of RNA sequencing data and bioinformatic support; Elisabeth J. Rushing for helpful advice, and Alexander Henzi for support with R programming and statistical analyses.

# Author Contributions

**Conceptualization:** Anna Henzi, Assunta Senatore, Asvin K. K. Lakkaraju, Claudia Scheckel, Jonas Mühle, Regina Reimann, Silvia Sorce, Gebhard Schertler, Klaus V. Toyka, Adriano Aguzzi.

**Data curation:** Anna Henzi, Assunta Senatore, Asvin K. K. Lakkaraju, Claudia Scheckel, Jonas Mühle, Regina Reimann, Silvia Sorce, Klaus V. Toyka, Adriano Aguzzi.

**Formal analysis:** Anna Henzi, Assunta Senatore, Asvin K. K. Lakkaraju, Claudia Scheckel, Jonas Mühle, Klaus V. Toyka.

**Funding acquisition:** Anna Henzi, Gebhard Schertler, Adriano Aguzzi.

**Investigation:** Anna Henzi, Assunta Senatore, Asvin K. K. Lakkaraju, Jonas Mühle, Regina Reimann, Silvia Sorce, Klaus V. Toyka.

**Methodology:** Anna Henzi, Assunta Senatore, Asvin K. K. Lakkaraju, Claudia Scheckel, Jonas Mühle, Regina Reimann, Silvia Sorce, Gebhard Schertler, Klaus V. Toyka, Adriano Aguzzi.

**Project administration:** Adriano Aguzzi.

**Resources:** Gebhard Schertler, Klaus V. Toyka, Adriano Aguzzi.

**Supervision:** Assunta Senatore, Asvin K. K. Lakkaraju, Gebhard Schertler, Adriano Aguzzi.

**Validation:** Anna Henzi, Assunta Senatore, Adriano Aguzzi.

**Visualization:** Anna Henzi.

**Writing – original draft:** Anna Henzi.

**Writing – review & editing:** Anna Henzi, Assunta Senatore, Asvin K. K. Lakkaraju, Claudia Scheckel, Jonas Mühle, Gebhard Schertler, Klaus V. Toyka, Adriano Aguzzi.

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
