## [Decision Letter · Decision Letter 0]

25 Sep 2020

PONE-D-20-23889

Soluble dimeric prion protein ligand activates Adgrg6 receptor but does not rescue the myelinopathy of PrP-deficient mice

PLOS ONE

Dear Dr. Aguzzi,

Thank you for submitting your manuscript to PLOS ONE. After careful consideration, we feel that it has merit but does not fully meet PLOS ONE’s publication criteria as it currently stands. Therefore, we invite you to submit a revised version of the manuscript that addresses the points raised during the review process.

We look forward to receiving your revised manuscript.

Kind regards,

Giuseppe Legname

Academic Editor

PLOS ONE

Journal Requirements:

2. To comply with PLOS ONE submissions requirements, please provide methods of sacrifice in the Methods section of your manuscript.

3.PLOS ONE now requires that authors provide the original uncropped and unadjusted images underlying all blot or gel results reported in a submission’s figures or Supporting Information files. This policy and the journal’s other requirements for blot/gel reporting and figure preparation are described in detail at https://journals.plos.org/plosone/s/figures#loc-blot-and-gel-reporting-requirements and https://journals.plos.org/plosone/s/figures#loc-preparing-figures-from-image-files. When you submit your revised manuscript, please ensure that your figures adhere fully to these guidelines and provide the original underlying images for all blot or gel data reported in your submission. See the following link for instructions on providing the original image data: https://journals.plos.org/plosone/s/figures#loc-original-images-for-blots-and-gels.

Reviewers' comments:

Reviewer's Responses to Questions

**Comments to the Author**

1. Is the manuscript technically sound, and do the data support the conclusions?

Reviewer #1: Yes

Reviewer #2: Yes

2. Has the statistical analysis been performed appropriately and rigorously? 

Reviewer #1: Yes

Reviewer #2: Yes

3. Have the authors made all data underlying the findings in their manuscript fully available?

Reviewer #1: Yes

Reviewer #2: Yes

4. Is the manuscript presented in an intelligible fashion and written in standard English?

Reviewer #1: Yes

Reviewer #2: Yes

5. Review Comments to the Author

Reviewer #1: Henzi and colleagues from the Aguzzi lab have designed a soluble dimeric version of the PrP flexible tail (FT2-Fc) that, like the flexible tail itself, is capable of binding and activating Adgrg6. Application of FT2-Fc to SW10 cells, but not SW10 cells lacking Adgrg6, resulted in a dose-dependent increase in cAMP levels and a time-dependent increase in phospho-AKT levels. This suggested that FT2-Fc may be capable of counteracting the age-dependent demyelinating phenotype that occurs in PrP knockout mice. However, neither acute nor persistent injections of FT2-Fc into mice resulted in an increase in phospho-AKT levels. Moreover, when PrP knockout mice were treated with FT2-Fc for 4 months, levels of Egr2 remained low and levels of Gfap remained high (compared to wild-type mice), suggesting that FT2-Fc was not engaging the appropriate signaling pathway. Yet, gene expression studies demonstrated alterations to the transcriptome in the sciatic nerves of FT2-Fc-treated mice, suggesting that the molecule reached its anatomical target. However, FT2-Fc treatment did not result in alterations to expression levels in genes relevant to Schwann cell repair. Instead, changes were observed in genes that undergo differential regulation upon PrP overexpression. These results suggest that FT2-Fc may not be a viable candidate for treating peripheral demyelinating disorders.

This is really nice, well-controlled study. The only major caveat to the conclusions, which is explicitly mentioned by the authors at the beginning of the Discussion, is that the mice were not treated long enough (or beginning at old enough ages) to directly test whether FT2-Fc rescues the peripheral nerve demyelination phenotype in PrP knockout mice. However, I do not fault the authors one bit for not doing these studies given the discouraging signaling pathway results from mice treated at younger ages. That being said, I think the authors should be careful with the wording of their conclusions in the Abstract and manuscript title. In the Abstract, it is stated that “no amelioration of the peripheral demyelinating neuropathy was detected.” A more accurate statement may be that “no amelioration of deficits in signaling pathways relevant to myelin maintenance was detected” (or something to that effect). The same goes for the title, since the authors did not actually test whether FT2-Fc “rescues the myelinopathy of PrP-deficient” mice. Otherwise, I think this is an excellent manuscript and have only a few other minor concerns for the authors to address:

1. Figure 3c clearly shows that Egr2 levels in ZH3 mice are lower than in wild-type mice. However, on page 16 it states in the Figure 3 legend that “ZH3 mice exhibited an age-dependent decrease in Egr2 levels” and in the associated main text that “Egr2 levels progressively decreased when compared to wild-type mice”. In my opinion, the Egr2 Western blot in Figure 3c does not clearly show a progressive decrease in levels with age in either the wild-type or ZH3 mice. The authors should quantify the Egr2 blot signals for these experiments, ideally from ~3 mice per age and genotype.

2. For Figure 2b, the authors need to briefly discuss why unexpectedly lower cAMP levels were obtained when using 9 or 10 µM FT2-Fc.

3. In the Methods section on page 5, I think one of the mouse lines should be CAG-/Cre+ (two are listed as CAG-/Cre- at the moment).

Reviewer #2: This manuscript describes studies aimed at evaluating the possible binding of an Fc-fusion protein (FT2Fc) comprising two units of the myelinotrophic domain of the prion protein, PrP, coupled to Fc, to the G-protein coupled receptor Adgrg6, resulting in its subsequent activation. Once such binding and activation were confirmed in cellula, animal experiments were carried out to evaluate whether FT2Fc might also have an effect in vivo. Such effect would consist in a partial reversal of the molecular events leading to the demyelination that takes place in absence of PrP. This work is a follow-up of a previous study by the Authors in which they demonstrated that the myelinotropic domain of PrP binds to and modulates Adgrg6, contributing to myelin maintenance (Kuffer et al., 2016; Nature 536(7617):464-8). The Authors reasoned that such effect might be used to our advantage to develop pharmacological modulators of Adgrg6 to treat demyelinating conditions. In this context, rather than evaluating the effect of the myelinotrophic domain of PrP, a short peptide that within PrP is known to be very flexible and is therefore likely to be disordered when alone, they decided to build the much more solid FT2Fc protein, expected to have better pharmacokinetic properties.

For their purposes, the Authors used a formidable array of molecular, cellular, histological, biophysical and electrophysiological techniques. Experiments were conducted rigorously, with adequate controls in every case; data were processed with appropriate statistical methods and are presented in a clear fashion. The results clearly show that FT2Fc activates Adgrg6 signaling in cellula, with the result of an increased cellular concentration of cAMP and increased AKT phosphorylation.

Unfortunately, a similar effect was not seen in animal studies. Rather, and paradoxically, changes in gene expression reminiscent of those detected in mice overexpressing PrP, and associated to toxicity, were seen in the sciatic nerves of PrP-deficient mice treated with intravenous or intraperitoneal injections of FT2Fc. Given the solidity of the analytical approach used to assess the effect of the treatment, it is possible to conclude that the treatment failed. While negative results are always disappointing, I think that the study will be of interest to the research communities studying the biology of the prion protein and maintenance of myelin, because of the questions it raises and the methodology the Authors have developed to tackle them. I will list a number of minor suggestions and queries, mostly related to the Discussion:

1) The Authors carried out pharmacokinetic studies based on a relatively simple approach: immunochemical detection of FT2Fc in serum. Did they consider using the same approach to measure FT2Fc in nerve tissue? This might be of relevance to determine if FT2Fc failed to elicit a response in vivo simply because it did not attain a sufficient presence in the target tissues.

2) The altered pattern of gene expression in the sciatic nerve induced by FT2Fc treatment was compared with the effect of overexpression of PrP in the tibialis muscle of tg mice. Why not use the pattern of gene expression in the sciatic muscle of these PrP overexpressing mice as a more relevant comparison?

3) The Authors suggest that "the dimeric nature of FT2Fc could alter its biological activity when compared to monomeric FT and endogenous PrP" as a possible explanation of its lack of effect (Discussion, line 603). However, the studies conducted in cellula showed that the effect of FT2Fc in activation of the signalling pathways considered was similar to that of monomeric FT.

4) What would be the mechanism by which FT2Fc exerts its PrP overexpression-like effect in gene expression? Would it be mediated by Adgrg6? If so, it means that pharmacological activation of this receptor might be catastrophic if not precisely fine-tuned; if mediated by other receptor, it means that, interestingly, FT has more than one binding partner/function.

6. PLOS authors have the option to publish the peer review history of their article (what does this mean?). If published, this will include your full peer review and any attached files.

Reviewer #1: No

Reviewer #2: **Yes: **Jesús R. Requena

---

## [Author Response · Author response to Decision Letter 0]

21 Oct 2020

We have formatted the manuscript according to PLOS ONE’s style requirements. We have also used PACE to convert our main figures to the accepted format of PLOS ONE.

2. To comply with PLOS ONE submissions requirements, please provide methods of sacrifice in the Methods section of your manuscript.

We have now provided the details in the Methods section (see p.6 of the revised manuscript with track changes).

3.PLOS ONE now requires that authors provide the original uncropped and unadjusted images underlying all blot or gel results reported in a submission’s figures or Supporting Information files. In your cover letter, please note whether your blot/gel image data are in Supporting Information or posted at a public data repository, provide the repository URL if relevant, and provide specific details as to which raw blot/gel images, if any, are not available. Email us at plosone@plos.org if you have any questions.

We have generated a single pdf file (S1_raw_images) comprising all the uncropped images of blots and gels. The images can be found in Supporting Information, which is also specified in the Methods section of the manuscript.

We are planning to upload the RNAseq data on NCBI GEO database. Unfortunately, these servers are currently down. We will provide the accession number and respective data availability statement as soon as possible.

Reviewer #1

I think the authors should be careful with the wording of their conclusions in the Abstract and manuscript title. In the Abstract, it is stated that “no amelioration of the peripheral demyelinating neuropathy was detected.” A more accurate statement may be that “no amelioration of deficits in signaling pathways relevant to myelin maintenance was detected” (or something to that effect). The same goes for the title, since the authors did not actually test whether FT2-Fc “rescues the myelinopathy of PrP-deficient” mice.

We agree with the reviewer that this is a major limitation of our study and we have adapted the wording in title, abstract and discussion of the manuscript to better reflect our findings.

1. Figure 3c clearly shows that Egr2 levels in ZH3 mice are lower than in wild-type mice. However, on page 16 it states in the Figure 3 legend that “ZH3 mice exhibited an age-dependent decrease in Egr2 levels” and in the associated main text that “Egr2 levels progressively decreased when compared to wild-type mice”. In my opinion, the Egr2 Western blot in Figure 3c does not clearly show a progressive decrease in levels with age in either the wild-type or ZH3 mice. The authors should quantify the Egr2 blot signals for these experiments, ideally from ~3 mice per age and genotype.

The reviewer has raised a valid point and it is indeed true that Figure 3c does not support the statement of a “progressive” decrease of Egr2 levels. We have now modified the text accordingly to reflect this in the respective results section and in the figure legend (see p.17). We have refrained from sacrificing more mice to do the quantifications as the precise changes in Erg2 levels do not alter the final conclusions of the experiment and the manuscript. Furthermore, we feel that such data would still not explain why the FT2Fc treatment failed to rescue the molecular signs of neuropathy. 

2. For Figure 2b, the authors need to briefly discuss why unexpectedly lower cAMP levels were obtained when using 9 or 10 µM FT2-Fc.

The unexpected shape of the dose-response curve might for example be related to receptor desensitization or to the dimeric nature of FT2Fc. Further studies will need to address the exact activation and desensitization mechanisms of Adgrg6. We included a short discussion about the unexpected dose-response curve in the Results (p.15) and Discussion sections (p.26) of the revised manuscript.

3. In the Methods section on page 5, I think one of the mouse lines should be CAG-/Cre+ (two are listed as CAG-/Cre- at the moment).

We thank the reviewer for highlighting this error. We have now corrected this and another mistake in the Methods section about the breeding of the PrP overexpressing mice, which is now described correctly (p.5).

Reviewer #2

1) The Authors carried out pharmacokinetic studies based on a relatively simple approach: immunochemical detection of FT2Fc in serum. Did they consider using the same approach to measure FT2Fc in nerve tissue? This might be of relevance to determine if FT2Fc failed to elicit a response in vivo simply because it did not attain a sufficient presence in the target tissues.

We were able to detect FT2Fc in nerve lysate by western blotting. We have added a supplementary figure showing the respective blot and included a brief discussion of these results in the manuscript (p.19, p.26). 

2) The altered pattern of gene expression in the sciatic nerve induced by FT2Fc treatment was compared with the effect of overexpression of PrP in the tibialis muscle of tg mice. Why not use the pattern of gene expression in the sciatic muscle of these PrP overexpressing mice as a more relevant comparison?

We assume that reviewer #2 meant “sciatic nerve of these PrP overexpressing mice”. The transgenic mice used in our study have only muscle-specific PrP overexpression. Thus, we do not expect significant gene expression changes in the peripheral nerves of these mice. We have extended the discussion about the myotoxicity of PrP-based agents to include the valid criticism raised by reviewer #2 (p.25-26).

3) The Authors suggest that "the dimeric nature of FT2Fc could alter its biological activity when compared to monomeric FT and endogenous PrP" as a possible explanation of its lack of effect (Discussion, line 603). However, the studies conducted in cellula showed that the effect of FT2Fc in activation of the signalling pathways considered was similar to that of monomeric FT.

We agree that FT2Fc works similarly to FT in the two in vitro cellular assays performed here. However, we observed a significantly higher potency of FT2Fc in the cAMP assay as well as an unexpected shape of the dose-response curve when compared FT. These results may point towards an altered biological activity when compared to FT. Furthermore, differences in bioavailability of dimeric proteins (e.g. half-life, avidity) may alter their biologic activity in in vivo experiments. This may be one reason why FT2Fc did not show the expected activity in vivo. We have addressed these concerns in the results and discussion sections.

4) What would be the mechanism by which FT2Fc exerts its PrP overexpression-like effect in gene expression? Would it be mediated by Adgrg6? If so, it means that pharmacological activation of this receptor might be catastrophic if not precisely fine-tuned; if mediated by other receptor, it means that, interestingly, FT has more than one binding partner/function.

Adgrg6 was previously suggested to be expressed in muscle tissue (Moriguchi, Genes Cells, 2004). However, another study (Musa, Ann. N.Y. Acad. Sci., 2019) objected that Adgrg6 detected in several tissues derived from expression in endothelial cells. Thus, we cannot be sure to date whether the detrimental function of PrP in muscle tissue is mediated by Adgrg6 or via other, unknown pathways. We are currently investigating the mechanism of PrP toxicity in skeletal muscle and we will consider both the possibility of Adgrg6-mediated and Adgrg6-independent effects. We have now discussed the questions raised by reviewer #2 in the manuscript (p.25-26).

---

## [Decision Letter · Decision Letter 1]

28 Oct 2020

Soluble dimeric prion protein ligand activates Adgrg6 receptor but does not rescue early signs of demyelination in PrP-deficient mice

PONE-D-20-23889R1

Dear Dr. Aguzzi,

We’re pleased to inform you that your manuscript has been judged scientifically suitable for publication and will be formally accepted for publication once it meets all outstanding technical requirements.

Kind regards,

Giuseppe Legname

Academic Editor

PLOS ONE

Additional Editor Comments (optional):

Reviewers' comments:

Reviewer's Responses to Questions

**Comments to the Author**

1. If the authors have adequately addressed your comments raised in a previous round of review and you feel that this manuscript is now acceptable for publication, you may indicate that here to bypass the “Comments to the Author” section, enter your conflict of interest statement in the “Confidential to Editor” section, and submit your "Accept" recommendation.

Reviewer #1: All comments have been addressed

Reviewer #2: All comments have been addressed

2. Is the manuscript technically sound, and do the data support the conclusions?

Reviewer #1: Yes

Reviewer #2: Yes

3. Has the statistical analysis been performed appropriately and rigorously? 

Reviewer #1: Yes

Reviewer #2: Yes

4. Have the authors made all data underlying the findings in their manuscript fully available?

Reviewer #1: Yes

Reviewer #2: Yes

5. Is the manuscript presented in an intelligible fashion and written in standard English?

Reviewer #1: Yes

Reviewer #2: Yes

6. Review Comments to the Author

Reviewer #1: (No Response)

Reviewer #2: (No Response)

7. PLOS authors have the option to publish the peer review history of their article (what does this mean?). If published, this will include your full peer review and any attached files.

Reviewer #1: No

Reviewer #2: **Yes: **Jesús R. Requena

---

## [Editor Report · Acceptance letter]

5 Nov 2020

PONE-D-20-23889R1 

Soluble dimeric prion protein ligand activates Adgrg6 receptor but does not rescue early signs of demyelination in PrP-deficient mice 

Dear Dr. Aguzzi:

I'm pleased to inform you that your manuscript has been deemed suitable for publication in PLOS ONE. Congratulations! Your manuscript is now with our production department. 

Kind regards, 

on behalf of

Prof. Giuseppe Legname 

Academic Editor

PLOS ONE